# Is Inflammation a Friend or Foe for Orthodontic Treatment?: Inflammation in Orthodontically Induced Inflammatory Root Resorption and Accelerating Tooth Movement

**DOI:** 10.3390/ijms22052388

**Published:** 2021-02-27

**Authors:** Masaru Yamaguchi, Shinichi Fukasawa

**Affiliations:** Ginza Orthodontic Clinic, Ginza Granvia 6F, 3-3-14 Ginza, Chuo-ku, Tokyo 104-0061, Japan; sinbikai@ginza.or.jp

**Keywords:** induced pluripotent stem cells, fabry disease, CRISPR/Cas9 gene editing, inflammatory

## Abstract

The aim of this paper is to provide a review on the role of inflammation in orthodontically induced inflammatory root resorption (OIIRR) and accelerating orthodontic tooth movement (AOTM) in orthodontic treatment. Orthodontic tooth movement (OTM) is stimulated by remodeling of the periodontal ligament (PDL) and alveolar bone. These remodeling activities and tooth displacement are involved in the occurrence of an inflammatory process in the periodontium, in response to orthodontic forces. Inflammatory mediators such as prostaglandins (PGs), interleukins (Ils; IL-1, -6, -17), the tumor necrosis factor (TNF)-α superfamily, and receptor activator of nuclear factor (RANK)/RANK ligand (RANKL)/osteoprotegerin (OPG) are increased in the PDL during OTM. OIIRR is one of the accidental symptoms, and inflammatory mediators have been detected in resorbed roots, PDL, and alveolar bone exposed to heavy orthodontic force. Therefore, these inflammatory mediators are involved with the occurrence of OIIRR during orthodontic tooth movement. On the contrary, regional accelerating phenomenon (RAP) occurs after fractures and surgery such as osteotomies or bone grafting, and bone healing is accelerated by increasing osteoclasts and osteoblasts. Recently, tooth movement after surgical procedures such as corticotomy, corticision, piezocision, and micro-osteoperforation might be accelerated by RAP, which increases the bone metabolism. Therefore, inflammation may be involved in accelerated OTM (AOTM). The knowledge of inflammation during orthodontic treatment could be used in preventing OIIRR and AOTM.

## 1. Introduction

Orthodontic tooth movement (OTM) is stimulated by remodeling of the periodontal ligament (PDL) and alveolar bone. These remodeling activities and tooth displacement are involved in the occurrence of an inflammatory process in the PDL and alveolar bone, in response to orthodontic forces. Cellular changes in PDL were the first events, and inflammatory mediators such as prostaglandins (PGs), interleukins (ILs; IL-1, -6, and -17), the tumor necrosis factor (TNF)-α superfamily, and the receptor activator of nuclear factor (RANK)/ RANK ligand (RANKL)/osteoprotegerin (OPG) increased in the periodontium. Their increased levels during OTM lead to biological responses that occur following the application of orthodontic forces [1].

Mechanical stimuli cause biological responses in a variety of cell types in the PDL and alveolar bone. The cellular mechanisms in response to tooth movement are very complex and their elucidation is important for investigators. In particular, investigation of inflammation during OTM focus on elucidation at the molecular level [2].

Many orthodontists consider orthodontically induced inflammatory root resorption (OIIRR) to be an unavoidable and unpredictably pathologic consequence of OTM. OIIRR is an iatrogenic disorder that occurs during orthodontic treatment, and the resorbed apical root portion is replaced with normal bone. The cause of OIIRR is unknown, although it is thought that complex inflammatory processes that involve various factors, such as mechanical forces, morphology of tooth roots, alveolar bone, PDL, cementum, and certain known biological messengers, are involved [3].

In 1959, Köle pointed out that corticotomies can accelerate tooth movement by cutting the alveolar bone [4,5,6]. Accelerated tooth movement is the result of the demineralization–mineralization process around the corticotomy, not the movement of the bone block. In 2001, this phenomenon was advocated as “Frost’s regional accelerating phenomenon (RAP) concept” by Wilcko et al. [7]. Wilcko et al. [8] also called it “bone matrix transportation”. Therefore, the inflammation caused by RAP may be involved in accelerating orthodontic tooth movement (AOTM).

In this article, we review current knowledge regarding inflammation in OIIRR and AOTM.

## 2. Optimal Orthodontic Force

According to Ren et al. [9] and Krishnan and Davidovitch [10], OTM is mediated by bone resorption and formation in the compressed and stretched sides of the PDL, respectively. Orthodontic force disturbs the homeostatic environment of the PDL space by the altering of blood flow and the localized environment. This alteration initiates biochemical and cellular events that maintain the alveolar bone conditions [11].

The concept of optimal force has changed considerably. Schwarz [12] proposed the classic concept of optimal force, and defined optimal force as “the force leading to a change in tissue pressure that approximated the capillary vessels’ blood pressure, thus preventing their occlusion in the compressed periodontal ligament.” Oppenheim [13] and Reitan [14] advocated the use of the lightest force for tooth movement.

The current concept of optimal force is that it is the mechanical force that leads to the maximum rate of tooth movement with minimal irreversible damage to the root, PDL, alveolar bone, and gingiva. The optimal force for tooth movement may differ for each tooth and for each individual patient [15].

## 3. Inflammatory Mediators in Orthodontic Tooth Movement

### 3.1. Prostaglandins (PGs)

According to Yamaguchi and Gaelet [16], PGs, a product of arachidonic acid metabolism, are local hormone-like chemical agents produced by mammalian cells, including osteoblasts, that are synthesized within seconds following cell injury. One of the derivatives of the arachidonic acid cascade, PGE2, acts as a vasodilator by causing increases in vascular permeability and chemotactic properties and it also stimulates the formation of osteoclasts and an increase in bone resorption. The cyclooxygenase (COX) family of enzymes consists of two proteins that convert arachidonic acid, a 20-carbon polyunsaturated fatty comprising a portion of the plasma membrane phospholipids of most cells, to PGs [17,18]. The constitutive isoform (COX-1) is found in nearly all tissues and is tissue protective. In contrast, COX-2, the inducible isoform of COX, appears to be limited in basal conditions within most tissues, and de novo synthesis is activated by cytokines, bacterial lipopolysaccharides), or growth factors to produce PGs in large amounts in inflammatory processes [19]. There are several lines of evidence showing that COX is also closely associated with periodontitis, and that PGs are mediators of gingival inflammation and alveolar bone resorption [20,21].

PGE2 stimulates the formation of osteoclasts and an increase in bone resorption. PGs are mediators of gingival inflammation and alveolar bone resorption [20,21]. The PGE2 levels in periodontal tissues and gingival crevicular fluid (GCF) in particular are highly correlated with periodontal tissue destruction [21]. Furthemorer, Lohinai et al. [22] reported that COX-2, the inducible isoform of cyclooxygenase (COX), is induced in gingival inflammation and alveolar bone destruction [22]. Ngan et al. [23] reported that the expression of PGs increased in the PDL and alveolar bone during orthodontic treatment. Shetty et al. [24] reported that the PGE2 levels in the GCF increased during OTM. Leiker et al. [25] demonstrated that exogenous prostaglandins enhanced the rate of OTM in rats. The administration of PGE or prostaglandin receptor EP4 also enhanced the rate of tooth movement [26,27]. Furthermore, indomethacin, a specific inhibitor of prostaglandin synthesis, reduces the rate of OTM in rats [27,28].

### 3.2. Cytokines

Cytokines are proteins that act as signals between the cells of the immune system, and are produced by inflammatory cells [2]. Cytokines such as IL-1, IL-6, IL-17, and TNF-α are the key mediators involved in a variety of immune and acute-phase inflammatory response activities [29,30,31]. 

IL-1 exists in two forms, α and β [32]; in particular, IL-β is involves with inflammation, and stimulates bone resorption [33,34,35]. Lo et al. [36] reported that IL-1β is produced by both macrophages and neutrophils, and is increased in inflamed gingival tissues. Kanda-Nakamura et al. [37] suggested that IL-1 may exacerbate gingival inflammation. 

IL-6 is a multifunctional cytokine produced by immune cells [38] and induces osteoclastic bone resorption [39,40,41]. In human gingival tissues, IL-6 is increased in inflammatory periodontal diseases [42]. Irwin and Myrillas [43] reported that IL-6 is involved to tissue destruction in the periodontal site. Yakovlev et al. [44] also reported that levels of IL-1β and IL-6 are increased in inflamed gingival tissues in young adults. In addition, Shimizu et al. [45] reported that IL-6 is stimulated by IL-1β from PDL cells in vitro.

IL-17 is an inflammatory cytokine that is produced by activated T cells (Th17 cells) [46]. IL-17 is an important mediator of autoimmune diseases, including rheumatoid arthritis (RA) [47], multiple sclerosis [48], and allergic airway inflammation [49]. Yago et al. reported that IL-17 induces osteoclastogenesis from monocytes [50]. In periodontal disease, IL-17 is related to bone destruction in periodontitis [47,51]. Furthermore, Honda et al. [52] demonstrated that the expression of IL-17 increases with the tissue destruction from periodontal diseases, and may be the Tcell-mediated pathogenesis of periodontal disease. 

TNF-α is also a pro-inflammatory cytokine that is involved in inflammatory diseases, including rheumatoid arthritis, inflammatory bowel disease [31], and periodontitis [53,54,55]. Takeichi et al. [56] reported that TNF-α is detected in periapical lesions with chronic periapical periodontitis and stimulates the initiation of inflammation and bone resorption. Gaspersic et al. [57] reported that TNF-α accelerates the progression of rat experimental periodontitis. Lee et al. [58] reported that TNF-α is increased in the GCF of periodontis and proceeds alveolar bone resorption. Therefore, there is a possibility that TNF-α is involved in osteoclastogenesis. The potential role of TNF-α in osteoclastogenesis is of significance in periodontitis. Ramadan et al. [59] put forward that inflammatory processes are accompanied by a large network of cytokines and chemokines in periodontitis. Cytokines including IL-17, IL-6, IL-1β, TNF-α, and PGE2, stimulate osteoclast activity, thereby causing bone resorption. 

Considering the in vitro and animal studies in the relationship between inflammatory cytokines and OTM, Saito et al. [60] demonstrated that IL-1 and PGE are involved in the response of periodontal cells to mechanical stress in vivo and in vitro. Baba et al. [61] and Vansant et al. [62] reported that OTM leads to the expression of the IL-1β gene in the PDL. Moreover, Zhang et al. [63] reported that that compressive force enhanced the expression of the IL-17 genes and osteoclastogenesis in osteoblasts in vitro. In addition, TNF-α is expressed in the PDL and alveolar bone during orthodontic tooth movement [64,65]. Moreover, in GCF studies during OTM, many reports have shown that PGE, IL-1, IL-6, IL-17, and TNF-α can be detected in compression and/or tension side [66,67,68,69]. Therefore, these inflammatory cytokines intricately intertwined with one another during OTM, and play important roles in bone remodeling. 

RANK ligand (RANKL, osteoclast differentiation factor, and osteoprotegerin ligand) and its receptor RANK are present on osteoblasts and precursor osteoclasts, respectively. They are the key factors that stimulate osteoclast formation and osteoclastogenesis [70,71]. RANKL is required for osteoclast formation with macrophage-colony stimulating factor (MCSF) from precursor monocyte/macrophages [72]. Osteoprotegerin (OPG), which a soluble TNF receptor-like molecule, inhibits RANK–RANKL interactions [73]. It binds to RANKL and prevents RANK–RANKL ligation. Therefore, OPG prevents osteoclast differentiation and activation. Animal studies using transgenic and gene knock-out mice have demonstrated that RANK, RANKL, and OPG play an important role in regulating physiologic osteoclast formation [74,75,76]. 

RANKL is produced by several types of cells in tissues adjacent to bone [77,78,79,80], and in inflammatory cells in inflamed tissues adjacent to areas of pathological bone loss in periodontal disease [77]. Liu et al. [81] and Ogasawara et al. [82] also reported that RANKL, in inflammatory cells, stimulates the activation of osteoclastic bone destruction in periodontitis. Therefore, RANKL is a powerful mediator of the progression of periodontal inflammation. Interestingly, the above-mentioned inflammatory cytokines and the RANK/RANKL/OPG system may stimulate bone resorption via regulation of the RANKL/OPG ratio. Previous studies have shown that IL-1, IL-11, IL-17, TNF, and PGE2 increases RANKL mRNA expression by T cells, while PGE2 decreases OPG expression [83,84]. De Molon reported that the gene and protein expressions of IL-1β, IL-6, and TNF-α are involved in osteoclastogenesis, as well as in the production of RANKL and OPG in experimental periodontitis [85].

Considering the relationship between RANKL/OPG and mechanical stress, Kanzaki et al. [86] demonstrated that compression forces up-regulate RANKL expression through induction of COX-2 in human PDL cells in vitro. Additionally, compression force increases RANKL and decreases OPG secretion in human PDL cells in vitro [87,88]. Animal studies [89,90] have demonstrated the expression of RANKL in periodontal tissues during rat experimental tooth movement. Furthermore, Kanzaki et al. [91,92] demonstrated that the amount of rat experimental tooth movement is accelerated by transfer of the RANKL gene to the periodontal tissue, while it is inhibited by OPG gene transfer. In GCF studies during OTM, the GCF levels of RANKL are increased, and the levels of OPG are decreased in experimental canine movement [93]. Therefore, it is suggested that the RANK–RANKL system is directly involved in the regulation of orthodontic tooth movement (Figure 1).

Accordingly, inflammatory cytokines, including IL-1β, IL-6, IL-17, and TNF-α, and the RANK–RANKL system play central roles in bone resorption during OTM (Table 1, Table 2 and Table 3).

## 4. Inflammation of Orthodontically Induced Inflammatory Root Resorption (OIIRR) and Inflammation—Is Inflammation a Foe?

Orthodontically induced inflammatory root resorption (OIIRR) induced by orthodontics is one of the inevitable complications of orthodontic tooth movement (Figure 2 and Figure 3). Factors related to orthodontic treatment and OIIRR—including prolonged treatment [94] and strong correctional force [95]—have been noted. Risk factors for the patient include genetic elements [96], age [97], root abnormalities [95], a history of tooth trauma [98], and allergies [99] (Figure 4).

Among the above, the applications of orthodontic forces induce inflammation around the PDL during OTM, and inflammatory factors are produced from PDL cells in response to orthodontic forces. PGs [93] and inflammatory cytokines [100,101,102,103] not only play a central role in bone resorption during OTM, but also in root resorption of the occurrence and exacerbation of OIIRR. 

In animal studies, Matsumoto et al. [104] reported that the expression of IL-1α, IL-1β, TNF-α, COX-2, and PGE2 in the resorbed root increases under heavy force during rat experimental tooth movement. Low et al. [105] reported that heavy force induces RANK and OPG in rat experimental tooth movement, which are involved with the occurrence of OIIRR. Furthermore, IL-6, IL-8, are IL-17 can be detected in resorbed root exposure to heavy forces in rat experimental tooth movement, and these cytokines may be involved in the occurrence of OIIRR [100,106,107] (Figure 5 and Figure 6).

In in vitro studies, Yamaguchi et al. [88] also reported that RANKL is produced by compression force and up-regulates osteoclastogenesis in vitro. Kikuta et al. [108] reported that excessive orthodontic forces stimulate the process of OIIRR via RANKL and IL-6 productions from hPDL cells. Tsukada et al. [109] reported that excessive compression force enhances TNF-a production from PDL cells. Therefore, the exposured of PDL cells to excessive compression force produces inflamatory cytokines and exacerbates the process of OIIRR. Interestingly, cementoblasts have also investigated the relationship between inflammatory cytokines and mechanical stress in vitro. Diercke et al. [103] demonstrated that IL-1β and compressive forces lead to a significant induction of RANKL expression in cementoblasts. Minato et al. [110] and Iwane et al. [111] reported that the exposure of cementoblasts to excessive orthodontic forces also produces a large amount of RANKL and IL-6, which induce the process of OIIRR. Therefore, the occurrence of OIIRR is caused by both PDL cells and cementoblasts.

From the above, as OIIRR may be due to a large amount of inflammatory cytokines from PDL cells and cementoblasts in response to heavy forces. Light forces may be recommended for minimizing the side effects. However, Yamaguchi and Garlet [16] suggested that heavy force may not be the most decisive factor for OIIRR, and that the severity of root resorption is highly dependent on how orthodontic force is applied. Acar et al. [112] and Maitha et al. [113] reported that intermittent forces cause less severe root resorption than continuous forces. Recently, jiggling force has become the focus as the cause of OIIRR. Jiggling forces produce more RANKL, IL-6, and IL-17 than in the heavy forces during rat experimental tooth movement [114,115,116]. These results suggest that jiggling force may be one of the risk factors for OIIRR. 

Yamaguchi and Garlet [16] made a new hypothesis about the relationship among inflammation, OTM, and OIIRR. It is possible that similar mechanisms operate in the “constructive” inflammation that mediates tooth movement, and in the “destructive” inflammation that results in root resorption. However, it is still unclear if differences in the intensity of the inflammatory process could explain the constructive/destructive dichotomy. Further studies are necessary to investigate this interesting hypothesis of OTM and OIIRR. From the above, OIIRR may be regulated by inflammatory cytokines, and inflammation may be a foe for orthodontic treatment.

In the 10 g group RANKL positive cells (black triangle) were observed in the bone resorption pit of the alveolar bone surface. In the 50 g group, RANKL positive cells (arrow heads) were observed in the root resorption pit. 

## 5. Accelerating Orthodontic Tooth Movement (AOTM) and Inflammation—Is Inflammation a Friend?

Many researchers and orthodontists have long attempted to reduce orthodontic treatment time. Clinical studies have reported that photobiomodulation [117], pharmacological approaches [118], and low-intensity laser irradiation [119] accelerate the rate of tooth movement. However, these stimulators still do not provide definite clinical effects. On the contrary, surgical procedures such as corticotomies are the most clinically applied and investigated [120]. 

RAP, a physiological reaction of reformation around the damaged area of bone, was referred to by Frost [121]. RAP is a reaction that occurs to heal the damaged area in the hard and soft tissues. In bone tissue, RAP increases bone turnover and decreases bone density to promote bone healing. These tissue responses vary depending on the duration, strength, and size of the harmful stimulus. Shih and Norrdin [122] demonstrated that RAP increases in bone remodeling in healing defects in beagle dogs with fractured ribs. Yaffe et al. [123] also reported that RAP that occurrs by a mucoperiosteal flap in the mandible of rats elevates born turnover.

Lee et al. [124] reported observations of demineralization/remineralization changes in the rat mandible by corticotomy, and that RAP occurrs at the operation site. RAP stimulates biological responses such as bone metabolism, bone cell differentiation, progenitor cell activity, growth of bone and cartilage, and bone remodeling by bone multicellular units [125,126]. RAP may also be caused by pharmacological stimuli including PGE1 and 2 and thyroxine [120,121,122]. In OTM, PGE1 and 2 increase the rate of OTM in animal experimental tooth movement [127]. In orthodontic treatment, Iino et al. [128] demonstrated a treatment duration of only one year in a 24-year-old Japanese women with bimaxillary protrusion treated by corticotomy. Therefore, tooth movement may be accelerated by RAP induced due to corticotomy, and as a result, treatment duration can be shortened compared to conventional orthodontic treatment.

In an earlier study in rats, Zhou et al. [129] demonstrated that corticotomy AOTM by increasing the gene expression of osteoblast-related factors (i.e., osteopontin, bone sialoprotein, and osteocalcin), osteoclast regulators (M-CSF, and RANKL). Furthermore, Zou et al. [130] reported that in rat experimental tooth movement, corticotomy + OTM increase a greater number of osteoclasts on the compression side, while increases osteocalcin on the tension side compared to OTM alone. Moreover, Sugimori et al. [131] reported that that micro-osteoperforation (MOPs) accelerates the rate of tooth movement via activation of the cell proliferation of PDL cells (Figure 7 and Figure 8). Furthermore, Yamaguchi et al. [132] reported that DNA chip analysis has demonstrated that the gene expression of minichromosome maintenance (MCM) and cell division cycle (CDC), related to cell proliferation, is increased by MOPs (Table 4). Therefore, inflammation stimulated by corticotomy/MOPs may induce the cell proliferation of PDL cells. Further studies will be necessary to clarify the relationship between inflammation and cell proliferation. Although corticotomy is an invasive procedure, the inflammation caused by the operation induces RAP. RAP reduces the resistance of bone during OTM, the stimulation of bone synthesis on the tension side, the activation of bone resorption on the compression side, and the stimulation of PDL metabolism, thereby shortening the period of orthodontic treatment and minimizing adverse effects on the teeth. Therefore, inflammation in surgical procedures may be a friend for orthodontic treatment.

A 29 year 2-month old Japanese woman presented with a chief complaint of protrusion of the maxillary incisors and upper and lower lips. The patient had no history of allergies or medical problems, and no signs or symptoms of temporomandibular dysfunction were observed.

Pretreatment facial photographs showed convex type and facial symmetry. The maxillary dental midline was shifted 0.5 mm to the right, compared to her facial midline. The patient had an Angle Class I malocclusion, with a 2.0 mm overjet and a 1.0 mm overbite. Both the maxillary and mandibular arches were irregularly aligned, with 2.0 mm maxillary and 0.5 mm mandibular arch length discrepancy (Figure 9). A panoramic radiograph showed four third molars. The lateral cephalometric analysis revealed the following characteristics: A normal skeletal relationship with an ANB angle of 0.5°, a moderate mandibular plane angle (FMA) of 24.0°, lingual inclination incisors with a maxillary central incisor to the Frankfort plane angle (FH-U1) of 126.5°, and a mandibular central incisor to the Frankfort plane angle (FMIA) of 52.0° (Figure 10 and Table 5). 

## 6. Diagnosis and Etiology

### 6.1. Diagnosis 

The patient was diagnosed with an Angle Class I malocclusion with bimaxillary protrusion.

### 6.2. Treatment Objectives

The main objective of the treatment of this malocclusion was to improve the protrusion of the maxillary and mandibular incisors and upper and lower lips.

### 6.3. Treatment Alternatives

This patient strongly requested orthodontic treatment with lingual appliances and desired a shortened treatment period. Therefore, we used corticision and piezocision of the maxillary and mandibular anterior teeth to shorten the active treatment period. In this case, the first premolars in the maxillary and the second premolars in the mandibular were extracted. 

### 6.4. Treatment Progress

The treatment was characterized by a combination of an upper lingual (Fukasawa Lingual Bracket (FLB), BIODENT, Japan) and lower labial (Empower, American Orthodontics, Sheboygan, WI, USA) appliance. Corticision and piezocision were planned after the appliance placements.

The corticision procedure was performed between the lower canine to canine areas. Subsequently, the patient underwent the piezocision procedure. In the corticision procedure, a scalpel (Blade No.15, Feather Safety Razor Co., Ltd., Osaka, Japan) was inserted into the anterior area to cut both the cortical bone and the cancellous bone. The scalpel was tapped with a mullet, following Park’s procedure. This procedure causes a surgical trauma at approximately 10 mm in depth, facilitating the next procedure. Next, a piezoelectric knife (Piezosurgery, Insert OT7, OT7S-4, MECTRON, Italy) was carefully inserted into the previously described area, following Dibart’s procedure.^11^ The purpose of this procedure was to stimulate the cancellous bone with a piezoelectric knife. Therefore, the depth of the piezoelectric knife insertion is critical in this procedure. Following extraction, the ostectomy procedure was performed to remove the alveolar bone bucco-lingually. In order to maintain the volume of the extraction sites, the removed bones can be used as autografts. Importantly, appropriate hemostasis procedures and medications are needed in the course of these surgical procedures.

Premolar extraction on both sides and the corticision and piezocision in the maxillary and the mandibular were archived (Figure 11), and leveling and alignment of the teeth were initiated with a preadjusted edgewise appliance (0.018 × 0.025 inches). In the present case, follow-up appointments took place every two weeks. Two months later, en masse retraction of the maxillary and mandibular anterior teeth was initiated. The extraction spaces were almost closed after six months of active treatment in the maxillary arch. The total active treatment period was only 16 months. The preadjusted edgewise appliance was removed and Hawley retainers were applied full time to the maxillary and mandibular arches, respectively. 

### 6.5. Treatment Results 

Owing to the successful retraction of the upper canines and anterior teeth, the patient showed an acceptable occlusion and good facial profile (i.e., balanced lip line). The dental arches were aligned and leveled, and an ideal overjet and overbite were achieved (Figure 12 and Figure 13). During active treatment after debonding, no significant periodontal problems (e.g., gingival recession or loss of tooth vitality) were observed (Figure 14). Pre- and post-treatment panoramic radiographs showed no significant reduction in the crest bone height and no marked apical root resorption (Figure 15). Additionally, pre- and post-treatment cephalometric superimpositions showed the absence of anchorage loss of the maxillary and mandibular molars. Furthermore, the interincisal angle (U1 to L1) was 125.0° post-treatment (Figure 16 and Table 5).

### 6.6. Case Summary

In this case of an adult patient with bimaxillary protrusion, the total treatment time was 16 months. This case report showed a correlation between orthodontic tooth movement and inflammation. Corticotomy shortens the orthodontic treatment period, including the extraction case by fixed lingual appliance, in adult patients. 

## 7. Conclusions

OTM causes inflammatory reactions in the periodontium. These reactions stimulate the release of various biochemical signals and mediators, causing alveolar bone and PDL remodeling. The control of inflammation and the efficiency of OTM are closely intertwined. This review demonstrated that inflammation causes OIIRR as a foe and AOTM as a friend. Therefore, orthodontists should treat patients while bearing in mind which inflammation will become a friend and a foe.

## Figures and Tables

**Figure 1 ijms-22-02388-f001:**
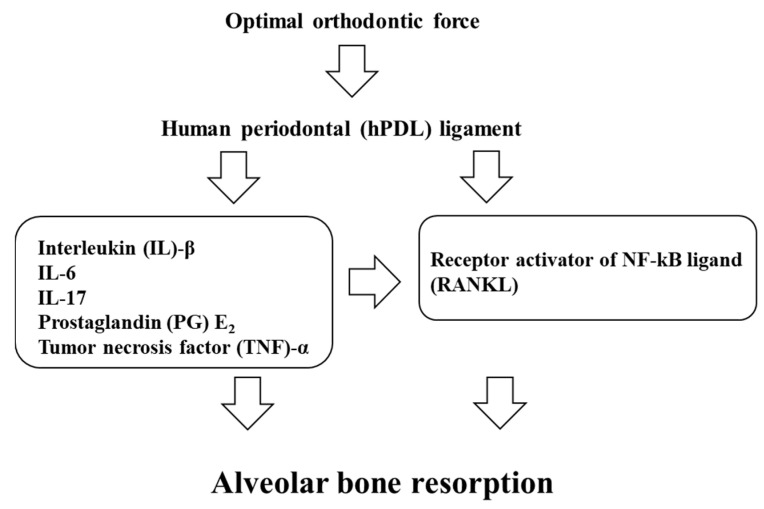
The schema of periodontal responses within the PDL on the pressure side.

**Figure 2 ijms-22-02388-f002:**
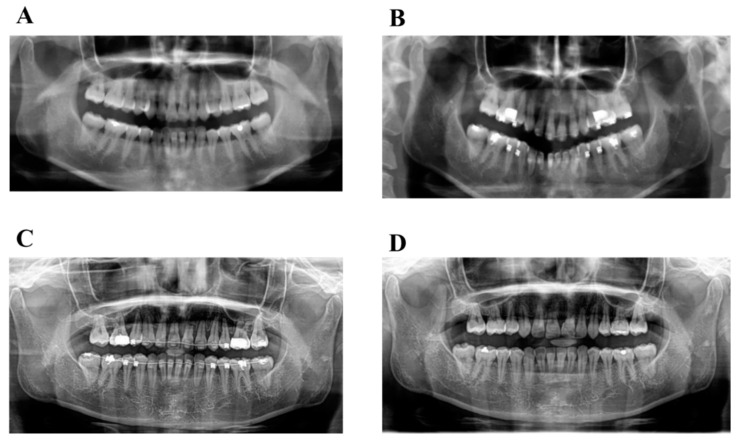
The progress panoramic radiograph of severe root resorption. (**A**): Pretreatment, (**B**): 1 year after start active treatment, (**C**): 2 years after start active treatment, (**D**): Post-treatment (2.5 years after start active treatment).

**Figure 3 ijms-22-02388-f003:**
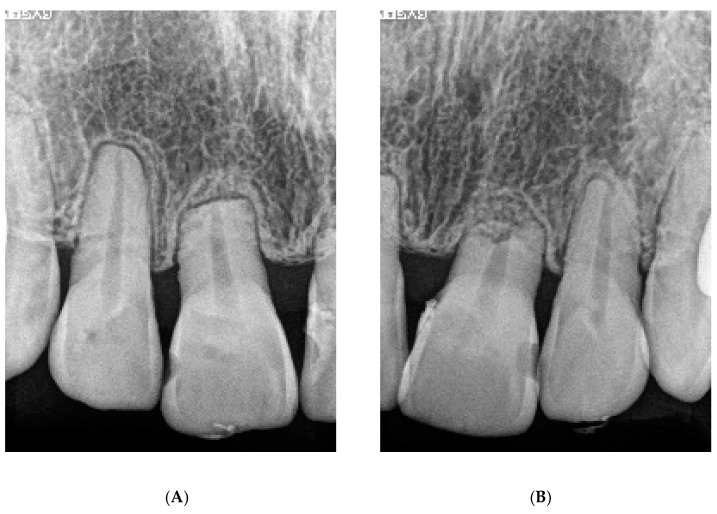
The dental radiograph of severe root resorption of Figure 2D (**A**): Right side, (**B**): Left side).

**Figure 4 ijms-22-02388-f004:**
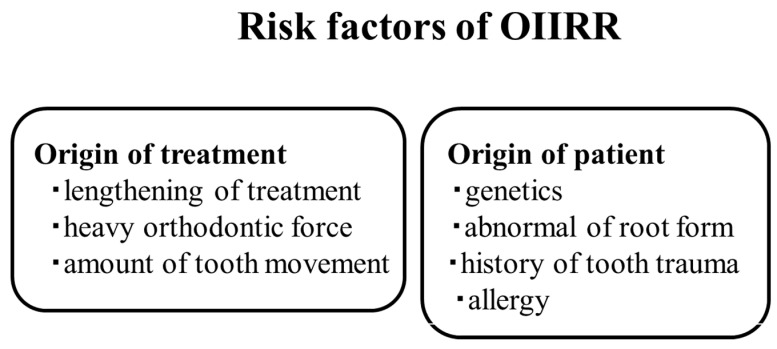
Risk factors of OIIRR.

**Figure 5 ijms-22-02388-f005:**
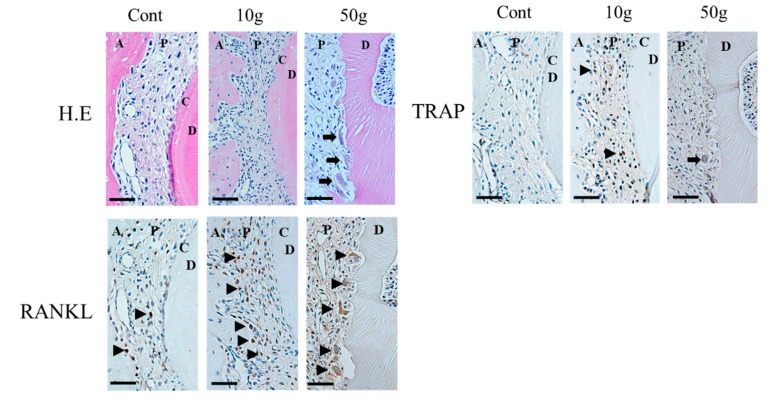
Histopathological staining (Hematoxylin-Eosin (H.E) staining), tartrate-resistant acid phosphatase (TRAP), and Receptor activator of nuclear factor kappa-B ligand (RANKL) immunohistochemical staining during rat experimental tooth movement after 7 days (performed by Yamaguchi M). In the 10 g group after 7 days, RANKL positive cells were observed in the bone resorption pit of the alveolar bone surface. In the 50 g group after 7 days, RANKL positive cells were observed in the root resorption pit. Bars: 50 μm. A, alveolar bone; P, periodontal ligament; C, cementum; D, dentin.

**Figure 6 ijms-22-02388-f006:**
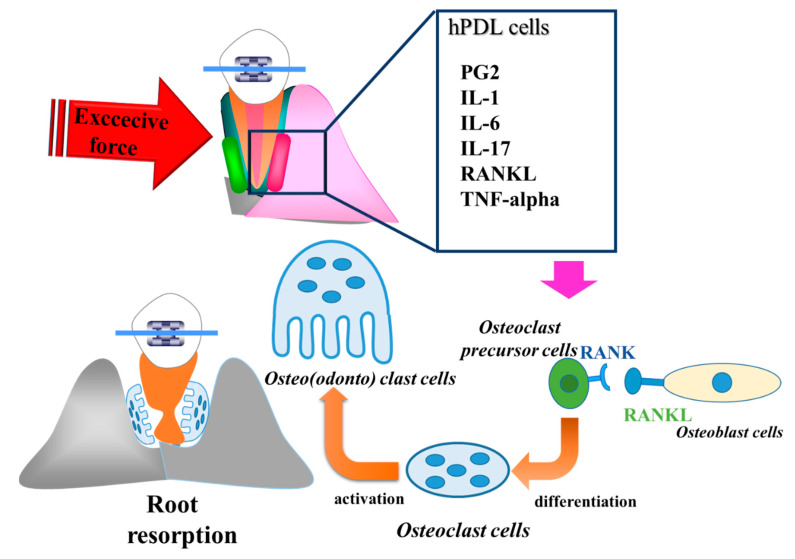
The schema of mechanism of OIIRR. Abbreviations; OIIRR = orthodontically induced inflammatory root resorption, hPDL cells = Human periodontal ligament cells, PGE2 = prostaglandin E2, IL-1 = interleukin 1, IL-6 = interleukin 6, IL-17 = interleukin 17, RANKL = Receptor activator of nuclear factor ligand, RANK = Receptor activator of nuclear factor, TNF-alpha = tumor necrosis factor-alpha.

**Figure 7 ijms-22-02388-f007:**
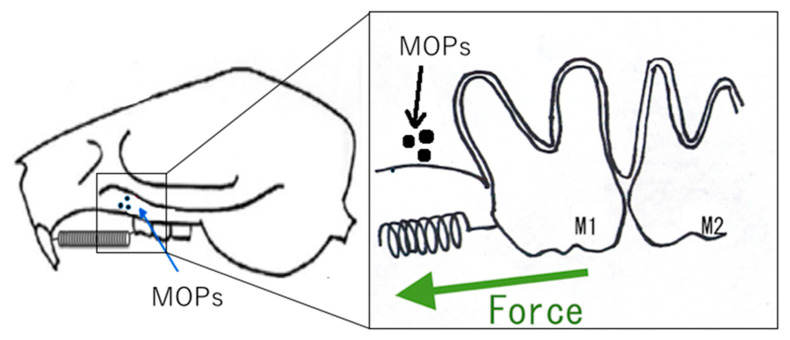
The schema of AOTM in rat experimental tooth movement model by MOPs [132]. Abbreviations: AOTM = accelerating orthodontic tooth movement, MOPs = micro-osteoperforations.

**Figure 8 ijms-22-02388-f008:**
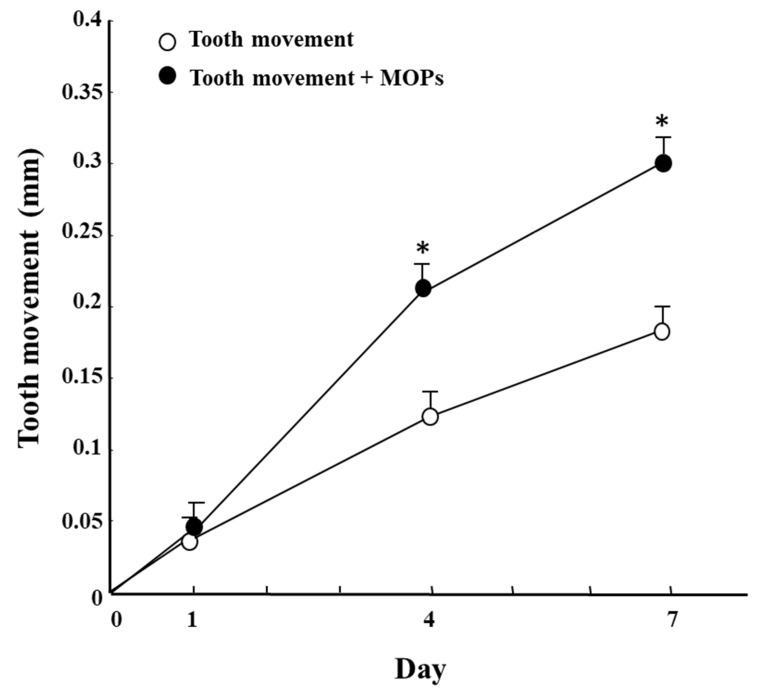
Effect of MOPs on tooth movement (ref. 132). * Significantly different from corresponding the tooth movement (TM) group and the TM+MOPs group (*p* < 0.05). Values are shown as the mean ± SD of 5 rats.

**Figure 9 ijms-22-02388-f009:**
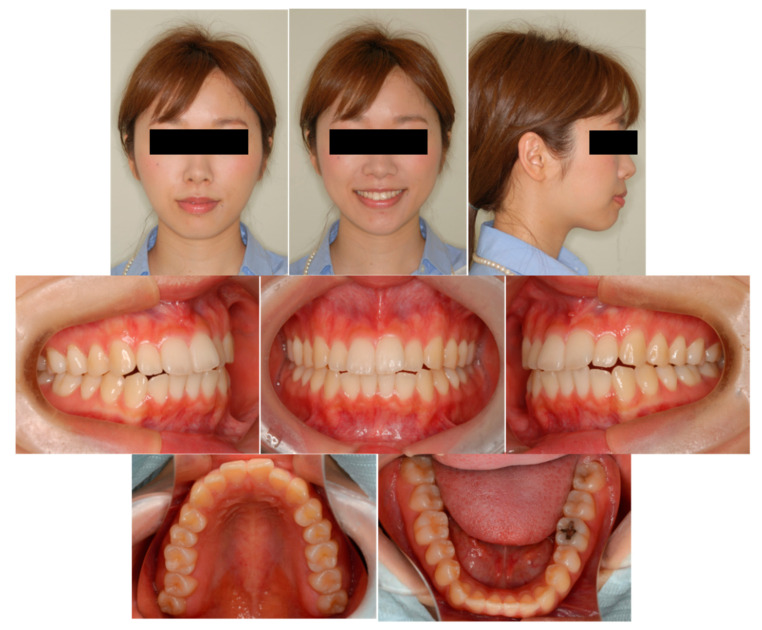
Pretreatment facial and intraoral photographs. (Patient is 29 years 2 months old).

**Figure 10 ijms-22-02388-f010:**
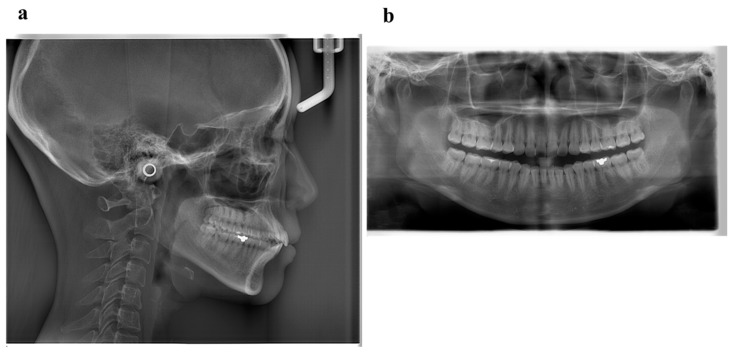
Pretreatment radiographs (**a**), lateral cephalogram; (**b**), panoramic radiograph).

**Figure 11 ijms-22-02388-f011:**
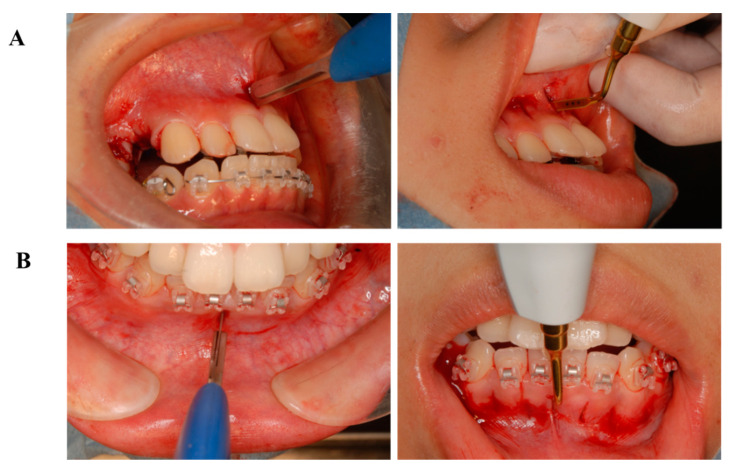
Progress intraoral photographs of corticision and piezocision (**A**), Maxilla; (**B**), Mandibular). The corticision procedure was performed between the lower canine to canine areas. Subsequently, the patient underwent the piezocision procedure.

**Figure 12 ijms-22-02388-f012:**
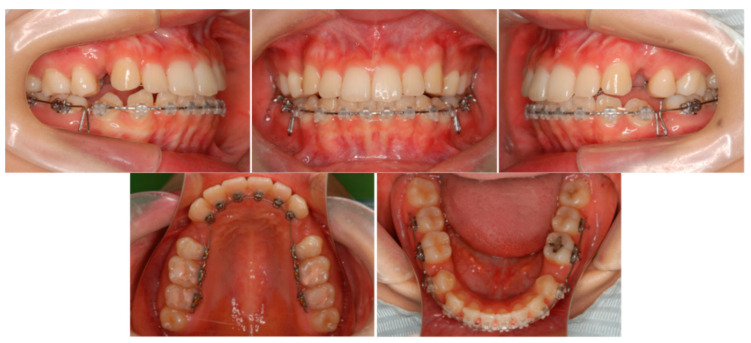
Progress intraoral photographs after 2 months of start of active treatment.

**Figure 13 ijms-22-02388-f013:**
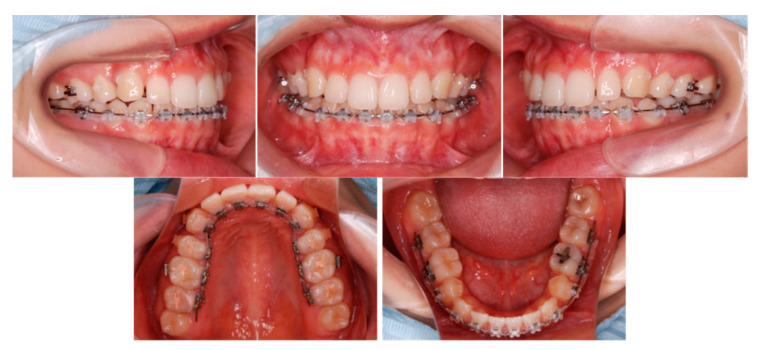
Progress intraoral photographs after 11 months of start of active treatment.

**Figure 14 ijms-22-02388-f014:**
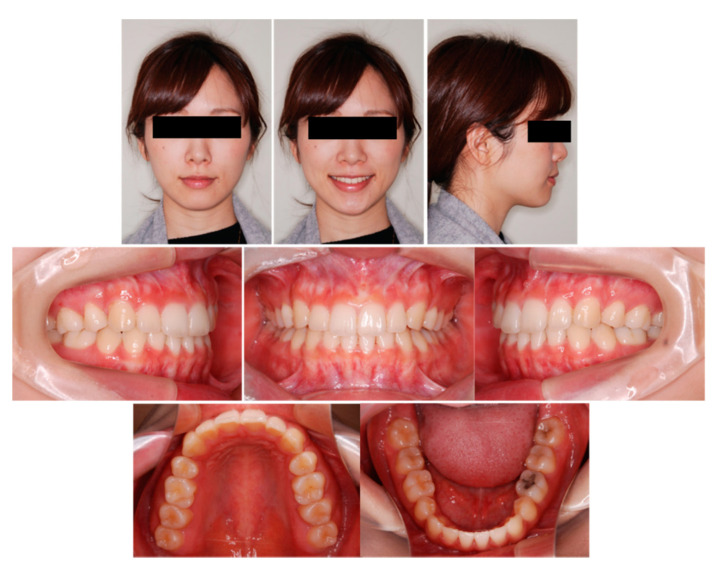
Post-treatment facial and intraoral photographs (after 16 months of start of active treatment; 31 years 2 months old).

**Figure 15 ijms-22-02388-f015:**
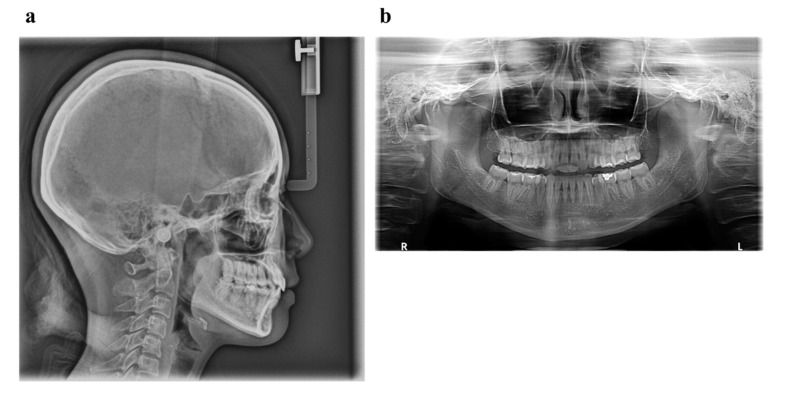
Post treatment radiographs (**a**), lateral cephalogram; (**b**), panoramic radiograph).

**Figure 16 ijms-22-02388-f016:**
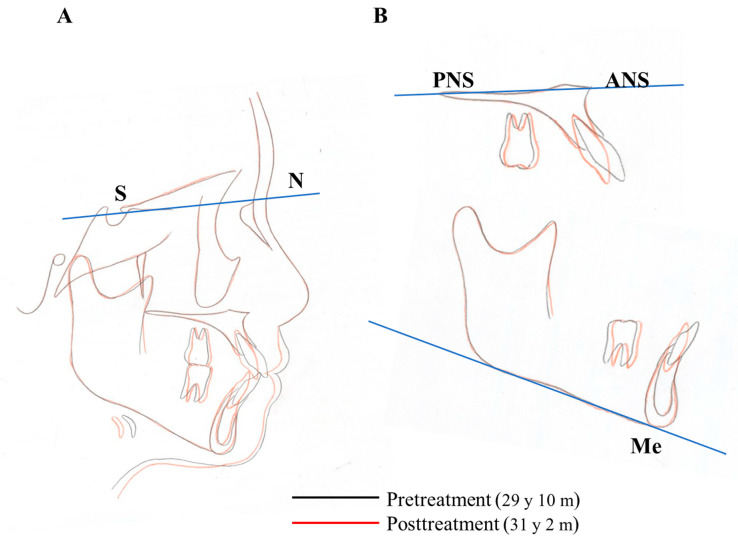
Superimposition of the cephalometric tracings at pretreatment and posttreatment. (**A**), The sella-nasion plane at sella; (**B**), Cephalometric superimpositions. Palatal plane at ANS and Mandibular plane at Me.

**Table 1 ijms-22-02388-t001:** In vitro studies in inflammation and mechanical stress.

Mechanical Stress	Cell	Inflammatory Mediator	Ref.
Compression force	Human PDL cells	IL-1, PGE	[60]
Compression force	Human PDL cells	IL-6	[94]
Compression force	MC3T3-E1 cells	IL-17	[63]
Compression force	Human PDL cells	RANKL/OPG	[86]
Compression force	Human PDL cells	RANKL/OPG	[87]

**Table 2 ijms-22-02388-t002:** Animal studies in inflammation and experimental tooth movement.

Type of TM	Animal	Inflammatory Mediator	Ref.
Canin Tip	Cat	IL-1, PGE	[60]
Molar Tip	Rat	TNF-α, IL-1β	[61]
Molar Tip	Rat	TNF-α, IL-1α	[64]
Rapid maxillary expansion	Human	TNF-α, RANKL	[65]
Molar Tip	Rat	RANKL/OPG	[89]
Human periapical granulomas	Human	RANKL/OPG	[90]

**Table 3 ijms-22-02388-t003:** Human GCF studies in inflammation and orthodontic tooth movement.

Tooth	Collection Side	Infammatory Mediator	Ref.
Canine	Pressure	PGE2, IL-1β	[66]
Canine	Pressure	IL-1β, IL-6, TNF-α	[67]
Canine	Pressure	IL-6, TNF-α	[68]
First premolar	Tension and pressure	IL-17	[69]
Canine	Pressure	RANKL/OPG	[87]

**Table 4 ijms-22-02388-t004:** DNA Chip analysis for MOPs during rat experimental tooth movement. Increasing minichromosome maintenance (MCM) and cell division cycle (CDC) gene expressions by MOPs [132].

Gene Symbol	Change (Fold)
CDC23	32.2
CDC7	12.1
MCM5	52.3
MCM7	19.0

**Table 5 ijms-22-02388-t005:** Cephalometric analysis before treatment and post-treatment.

Measurement	Normal (Japanese Woman)	Pretreatment (29 y 19 m)	Posttreatment (31y 2m)
SNA (° )	81.3 ± 2.7	80.0	80.0
SNB (° )	78.6 ± 2.7	79.5	79.0
ANB (° )	2.6 ± 1.1	0.5	1.0
FMA (° )	26.3 ± 4.1	24.0	24.0
FMIA (° )	56.9 ± 6.4	52.0	61.0
IMPA (° )	96.8 ± 6.4	104.0	95.0
U1-FH (° )	112.1 ± 4.2	126.5	115.5
U1-L1 (° )	123.5 ± 5.5	105.0	125.0
U1-A-Pog (mm)	6.2 ± 1.5	11.2	4.6
L1-A-Pog (mm)	3.0 ± 1.5	8.1	2.3
Gonial angle (° )	118.8 ± 6.1	117.5	117.5
E-line: Upper (mm)	1.4 ± 2.0	−0.5	−2.0
E-line: Lower (mm)	1.4 ± 2.0	+0.3	−0.7

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
