# Peer review of "Is Inflammation a Friend or Foe for Orthodontic Treatment?: Inflammation in Orthodontically Induced Inflammatory Root Resorption and Accelerating Tooth Movement"

_ijms, 2021, doi:10.3390/ijms22052388_

Round 1

Reviewer 1 Report

i found the extensive use of abbreviated letters to represent entire words was very confusing and difficult to remember what was being referred to as there were so many  .

The content of the article  will be most beneficial for graduate students to better understand the pathophysiology of tooth movement. 

Author Response

Response to Reviewer 1 Comments

Thank you for giving us valuable advice and comments regarding our manuscript. We revise upon the manuscript as follows.

Reviewer 1

Point 1: i found the extensive use of abbreviated letters to represent entire words was very confusing and difficult to remember what was being referred to as there were so many.

Response 1: We added the list of abbreviations in the text and Figures.

The content of the article will be most beneficial for graduate students to better understand the pathophysiology of tooth movement.

Response 2: We are honored by your kind words.

Reviewer 2 Report

The topic is of interest as inflammatory root resorption occurs frequently during and after orthodontic therapy. The major problem I see is that the manuscript contains many orthographical and grammatical errors. Following, it is difficult to read. I strongly recommend professional language editing.

Further I miss in part the focus on orthodontic therapy. Many citations are related to periodontal therapy. I suggest a better structuring presenting results of in-vitro and animal studies in tables.

Figures were included. Do they present own research? This should be clarified.

Table 1: The title should be self-explanatory without reading the text. And again, who made the research?

Finally the authors present in extenso a case which is certainly of interest to present but not at the end of a very long review.

Taken together, I recommend a total rewriting of the manuscript.

Author Response

Response to Reviewer 2 Comments

Thank you for giving us valuable advice and comments regarding our manuscript. We revise upon the manuscript as follows.

Reviewer 2

The topic is of interest as inflammatory root resorption occurs frequently during and after orthodontic therapy.

Point 1: The major problem I see is that the manuscript contains many orthographical and grammatical errors. Following, it is difficult to read. I strongly recommend professional language editing.

Response 1: As you indicated, this text was received professional English editing by MDPI English editing.

Point 2: Further I miss in part the focus on orthodontic therapy. Many citations are related to periodontal therapy. I suggest a better structuring presenting results of in-vitro and animal studies in tables.

Response 2: As you advised, the results of in-vitro and animal studies presented in tables 1.

Point 3: Figures were included. Do they present own research? This should be clarified.

Response 3: This data was the results of our research. It was clarified.

Point 4: Table 1: The title should be self-explanatory without reading the text. And again, who made the research?

Response 4: As you advised, the title was corrected and it was clarified as our research.

Point 5: Finally the authors present in extenso a case which is certainly of interest to present but not at the end of a very long review.

Response 5: Your opinion is justified. However, we want to emphasize specifically the relationship between inflammation and accelerating tooth movement. Therefore, the case report was presented in this review, and this review has become long.

Point 6: Taken together, I recommend a total rewriting of the manuscript.

Response 6: As you indicated, this text was received professional English editing by MDPI English editing.

Reviewer 3 Report

There are many typo in the manuscript, please check.

In 3.1. What is the difference between “Prostaglandins (PGs)” and “PGE2”?

This paragraph is introducing PGs, why COX-2 is mentioned here?

“IL-6 stimulatd by IL-1 from the human foreskin fibroblast in vitro” causes confusion.

Since this is a review it is better not be a case report.

Author Response

Response to Reviewer 3 Comments

Thank you for giving us valuable advice and comments regarding our manuscript. We revise upon the manuscript as follows.

Reviewer 3

Point 1: There are many typo in the manuscript, please check.

Response 1: As you indicated, this text was received professional English editing by MDPI. English editing.

Point 2: In 3.1. What is the difference between “Prostaglandins (PGs)” and “PGE2”?

Point 3: This paragraph is introducing PGs, why COX-2 is mentioned here?

Responses 2 and 3: We revised to clarify the relationship among PGE, PGE2, and COX-2.

Point 4: “IL-6 stimulatd by IL-1 from the human foreskin fibroblast in vitro” causes confusion.

Response 4: As you advised, this sentence was deleted.

Point 5: Since this is a review it is better not be a case report.

Response 5: Your opinion is justified. However, we want to emphasize specifically the relationship between inflammation and accelerating tooth movement. Therefore, the case report was presented in this review.

Round 2

Reviewer 1 Report

page 12--spelling of word excessive  

Author Response

Response to Reviewer 1 Comments

Thank you for giving us valuable advice and comments regarding our manuscript. We revise upon the manuscript as follows.

Reviewer 1

Point 1: page 12--spelling of word excessive 

Response 1: As you advised, these sentences was shortened. 

The staining of IL-6 was deleted.

Reviewer 2 Report

Table 1: Please give more Information on the results in the presented studies. 

I still recommend shortening the case report. Please clearly refer also in figure legends to results published before.

Author Response

Response to Reviewer 2 Comments

Thank you for giving us valuable advice and comments regarding our manuscript. We revise upon the manuscript as follows.

Reviewer 2

The topic is of interest as inflammatory root resorption occurs frequently during and after orthodontic therapy.

Point 1: Table 1: Please give more Information on the results in the presented studies.

Response 1: As you advised, more information was added in Tables 1, 2, and 3.

Point 2: I still recommend shortening the case report. Please clearly refer also in figure legends to results published before.

Response 2: The case report was shortened, and figure legends were revised clearly.

Reviewer 3 Report

  1. Too many mistakes in the manuscript.
  2. It looks like a mixture of review, research, and case report.
  3. What is “morpholigy of tooth roots”?
  4. What is “intericately interwinde”?
  5. There are many mistakes in “Osteoprotegerin (OPG), which a soluble TNF receptor-like molecule, inhibits RANK–RANKL interactions [76]. It binds to RANKL and prevents RANK–RANKL ligation. Therefore, OPG prevents osteoclast differentiation and activation. Animal studies using ransgenic and gene knock-out mice have demonstrated that RANK, RANKL, and OPG play am important role in regulating physiologic osteoclast formation [77-79].”.
  6. In Fig. 1 There are mistake in “periodontal ligament (hPDL” and “Tumor necrosis factor (TNF)-alfa”.
  7. What is the aim of Table 1? It would be better to list the relationship between cytokines and the outcome of OTM.

Author Response

Response to Reviewer 3 Comments

Thank you for giving us valuable advice and comments regarding our manuscript. We revise upon the manuscript as follows.

Reviewer 3

Point 1: Too many mistakes in the manuscript.

Response 1: I am terribly sorry. We corrected mistakes in the text.

Point 2: It looks like a mixture of review, research, and case report.

Responses 2 : The review is a special format that includes research and case reports. This is to help readers understand OIIRR and AOTM better. Therefore, the review includes research and case reports.

Point 3: What is “morpholigy of tooth roots”?

Responses 3: As you indicated, this mistake was corrected to “morphology of tooth roots”.

Point 4: What is “intericately interwinde”?

Response 4: As you indicated, this mistake was corrected to “intricately intertwined”.

.

Point 5: There are many mistakes in “Osteoprotegerin (OPG), which a soluble TNF receptor-like molecule, inhibits RANK–RANKL interactions [76]. It binds to RANKL and prevents RANK–RANKL ligation. Therefore, OPG prevents osteoclast differentiation and activation. Animal studies using ransgenic and gene knock-out mice have demonstrated that RANK, RANKL, and OPG play am important role in regulating physiologic osteoclast formation [77-79].”.

Response 5: As you indicated, this mistake was corrected to “transgenic” and “an”.

Point 6: In Fig. 1 There are mistake in “periodontal ligament (hPDL” and “Tumor necrosis factor (TNF)-alfa”.

Response 6: As you indicated, these mistakes were corrected.

Point 7: What is the aim of Table 1? It would be better to list the relationship between cytokines and the outcome of OTM.

Response 7: As you advised, more information of the relationship between cytokines and the outcome of OTM was added in Tables 1, 2, and 3.

This manuscript is a resubmission of an earlier submission. The following is a list of the peer review reports and author responses from that submission.